# Cortical PV and VIP interneurons similarly influence SST neuron output despite distinct unitary properties
Felix Preuss [1,2], Martin Möck[1], Mirko Witte[1] & Jochen F. Staiger [1] ✉

Somatostatin (SST) expressing cells are powerful inhibitors of excitatory pyramidal neurons. It has been shown that SST cells are targeted by other inhibitory interneurons, namely parvalbumin- (PV) and vasoactive intestinal polypeptide- (VIP) expressing cells in various cortical regions. Subcellular distribution of PV and VIP synapses suggests differences in their modulation of action potential generation in postsynaptic SST cells. However, functional analyses if and to what extent individual neurons are able to change the output of postsynaptic targets are sparse. To test this, we use paired patch clamp recordings to analyze SST cell firing with and without presynaptic PV or VIP cell stimulation. Despite their enormous differences in unitary synaptic properties, individual PV and VIP cells both are able to significantly decrease action potential output in postsynaptic SST cells. However, testing two different action potential firing durations (1 s and 100 ms) in presynaptic cells, we do not observe significant differences in overall spike loss of PV to SST versus VIP to SST cell connections. Morphological analysis of putative contact sites (PCS) does not reveal differences in PCS location. We propose that individual GABAergic neurons are indeed able to modulate the firing output of SST neurons without principled cell type specificity.

Although inhibitory GABAergic interneurons (IN) make up only ≈15% of all neurons in the rodent neocortex, they are an extensive field of study in contemporary neuroscience[1–6]. They can be subdivided into four non-overlapping groups based on the expression of different molecular markers: parvalbumin (PV), somatostatin- (SST), vasoactive intestinal peptide- (VIP) expressing cells and other cells that express the markers like Lamp5, Sncg, or Serpinf1[7–10]. These groups can be further subdivided into different subgroups and potential cell types[1,11,12].

GABAergic IN not only target excitatory cells but also other inhibitory cells, a circuit motif which is known as disinhibition[13]. The most prominent cortical disinhibitory circuit motif is the VIP to SST cell motif which has been described in multiple cortical areas and is thought to be involved in various behaviors[14–21]. SST-expressing Martinotti cells (MC) are characterized by an axon ascending into layer 1 (L1) and target the dendrites of excitatory pyramidal neurons (PN)[22–24]. By targeting SST cells, VIP cells are effectively able to increase PN activity[15]. Although PV cells are best-known for targeting excitatory cells[25–29], their disinhibitory role has received increased attention in the recent years[13,14,30,31]. Many studies that focus on inhibition mostly analyze connection probabilities and inhibitory post-synaptic currents (IPSCs). This allows an estimation of input strength, but

what this implies for the output of postsynaptic target cells remains speculative. Moreover, studies that functionally analyzed the disinhibitory PV/VIP to SST to PN circuit often used optogenetics to stimulate a larger population of presynaptic inhibitory cells[32–37] and only few studies analyzed the effect of individual presynaptic cells[38,39].

In theory, there are two mechanisms how individual IN are able to decrease firing in postsynaptic cells. Either by reduction of excitatory drive on postsynaptic dendrites (also known as shunting inhibition[40]) or by direct prevention of action potential generation in postsynaptic cells by sufficient hyperpolarization. Due to the perisomatic targeting of PV cells[41–43], they are thought to inhibit action potential generation in postsynaptic target cells (i.e., output control) while dendrite targeting VIP cells[44–46] are thought to decrease the excitatory drive via input control. We performed paired patch clamp recordings of L2/3 PV to SST and VIP to SST cell pairs in mouse barrel cortex in acute brain slices. Using somatic current injection, we tested whether individual presynaptic IN are able to change the output of their postsynaptic target cell. If the main role of VIP cells is indeed the reduction of excitatory (dendritic) drive, then only a marginal effect on action potential firing is to be expected. However, both cell types were able to reduce action potential generation in postsynaptic SST target cells. Our data question the

[1]Institute for Neuroanatomy, Universitätsmedizin Göttingen, Georg-August-Universität, Göttingen, Germany. [2]Neurophysics Laboratory, Göttingen Campus Institute for Dynamics of Biological Networks (CIDBN), Göttingen, Germany. ✉e-mail: jochen.staiger@med.uni-goettingen.de

notion of a strict separation of input versus output control being mediated by different types of GABAergic IN in the cortical circuit.

## Results

We used two different mouse lines (PV-Cre//tdTomato//GIN and VIP-Cre//tdTomato//GIN) to analyze the effect of presynaptic individual PV or VIP cell firing on the output of SST cells in L2/3 of mouse barrel cortex. Cre-driver lines are an essential and widely used tool in modern neuroscience[47–49]. GIN stands for GFP-expressing inhibitory interneurons and in this mouse line, a subpopulation of SST cells expresses the green fluorescent protein eGFP[50,51]. In a previous study, we were able to show that in triple transgenic animals, GIN cells in L2/3 can all be classified as MC, which are targeted by PV and VIP cells[14]. Genetically labeled cells demonstrated cell type-specific morphological and electrophysiological properties[14]. In the latter study, we used a Cs-based internal solution with a holding potential of 0 mV to increase the driving force for inhibitory chloride currents. Here, we wanted to record physiological action potentials in both cells. Therefore, we had to use a K-gluconate based internal solution in pre- and postsynaptic cells.

Since most frequently VIP neurons target dendrites and not somas[44], we assumed low-amplitude inputs of VIP neurons to SST cells. Thus, we first had to test whether we would be able to detect small VIP cell inputs under these recording conditions.

We performed paired patch clamp recordings of putative presynaptic PV or VIP cells and postsynaptic SST cells (exemplar reconstructed cell pairs are show in Fig. 1A). Inducing high frequency firing in presynaptic cells right away (instead of single action potentials) allowed us to identify connected cell pairs with certainty although a K-gluconate-based solution was used. We found remarkably high connectivity rates in both circuit motifs amounting to 43.2% (19/44) in PV to SST and 43.8% (35/80) in VIP to SST cell pairs (Fig. 1B). In most cases, we also tested the reverse SST to PV/VIP connection (Fig. 1C). Here we found a connectivity rate of 65.1% (28/43) in SST to PV and a substantial 31.0% (22/71) in SST to VIP cell pairs. Moreover, 84.2% (16/19) of PV to SST cell pairs were reciprocally connected but only 32.3% (10/31) of VIP to SST cell pairs (Fig. 1D). All cells displayed cell type-specific firing patterns, known as fast spiking, continuous adapting, irregular spiking, or bursting (Fig. 1E, F[52]). Exemplar paired recordings of a PV to SST and a VIP to SST cell pair are shown in Fig. 1G, H.

### Presynaptic PV and VIP cell firing decreases postsynaptic SST cell output

Our "connection test" recordings clearly demonstrated that we are able to identify connected PV to SST and VIP to SST cell pairs using a "physiological" K-gluconate-based internal solution, in contrast to previous studies[14,30]. This allowed us to test the functional effect on SST cell output of presynaptic PV or VIP cells in identified cell pairs. We used one-second-long somatic current injection pulses (called "long pulse" from now on) to induce continuous firing in PV or VIP cells (in order to simulate a strong input) and in SST cells (in order to simulate an output independent from integration of excitatory inputs all over the dendritic tree). This ensured a high reproducibility of firing patterns and overall number of action potentials in individual traces. We alternately recorded one-second-long

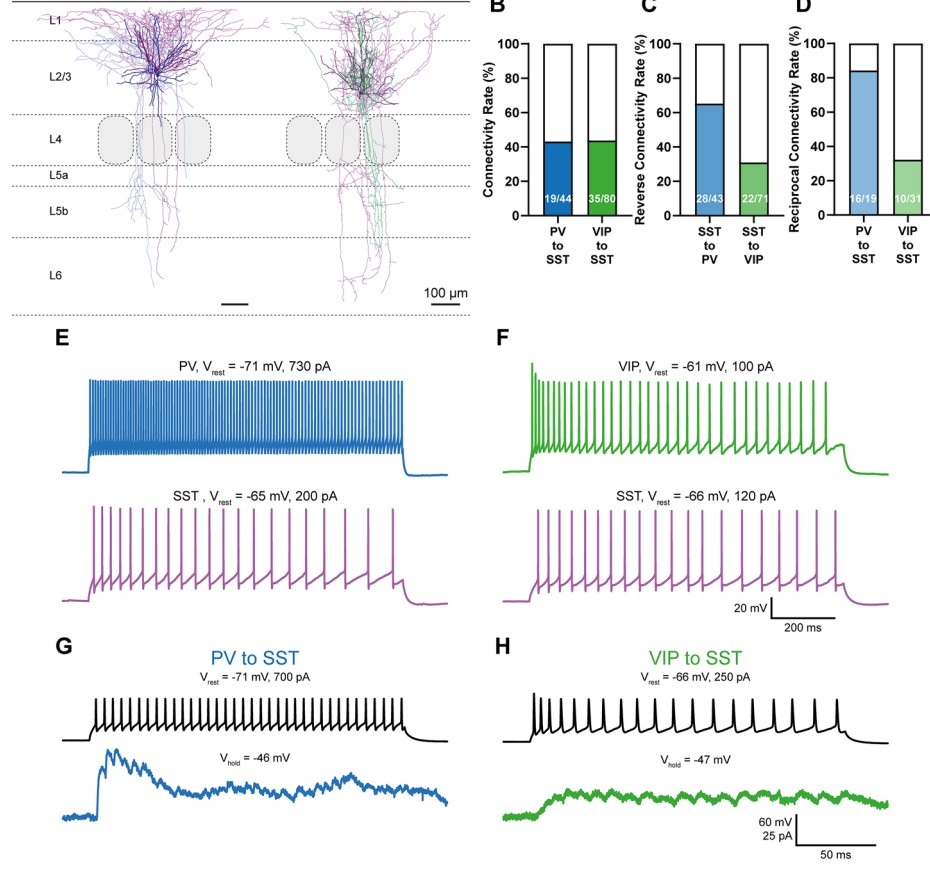

**Fig. 1 | PV and VIP cells effectively target SST cells and display cell type-specific differences.**
**A** Exemplar reconstructions of a connected PV to SST (left) and a VIP to SST cell pair (right) in L2/3 of mouse barrel cortex. SST soma and dendrites are shown in dark and axon in light purple. SST cells are of the MC type due their dense axonal projection to L1. PV cell soma and dendrites are shown in dark and axon in light blue. The PV cell can morphologically be described as BC. VIP cell soma and dendrites are shown in dark and axon in light green. Shown here is a bipolar VIP cell. Scale bar = 100 μm. **B–D** Connectivity rates of tested PV to SST and VIP to SST cell pairs (**B**) and reverse connectivity rates of SST to PV and SST to VIP connections (**C**). Especially in the PV to SST motif we observed high numbers of reciprocally connected cell pairs. Number of tested pairs is noted on each bar graph. In both directions, we observed high connectivity rates between both cell types. Exemplar recordings of action potential firing in a (presynaptic) PV and a (postsynaptic) SST cell (**E**) as well as a (presynaptic) VIP and another (postsynaptic) SST cell (**F**). Cells were held at resting membrane potential ($V_{rest}$). Action potential firing was induced by using one-second-long depolarizing current injection pulses. All cells displayed cell type-specific firing patterns. The PV cell displays a fast spiking whereas the VIP and both SST cells display a continuous adapting firing pattern. (G + H) Paired recording of cells shown in (**E**, **F**). Postsynaptic SST cells were held in voltage clamp, close to firing threshold ($\approx 50$ mV) to increase the driving force for inward chloride currents. Presynaptic PV and VIP cell firing led to IPSCs in postsynaptic SST cells (blue and green). We observed different short-term synaptic plasticity in both connections. The PV to SST connection was depressing, whereas we observed facilitation in the VIP to SST connection.

epochs of evoked SST cell firing alone (without stimulation of the presynaptic neurons) versus SST cell firing in conjunction with presynaptic PV or VIP cell stimulation (exemplar traces in Fig. 2A, B). Traces with presynaptic stimulation were always compared to previous traces without presynaptic stimulation.

We recorded and analyzed 15 PV to SST (age: P25-P48, median P42 from 14 animals) and 21 VIP to SST cell pairs (age: P28-P55, median P35, from 19 animals). Presynaptic stimulation of a single PV or VIP cell led to substantially decreased postsynaptic SST cell firing (median SST spike number without PV stimulation 29.80, with PV stimulation 25.50, $p = 0.0007$, paired t test; median SST spike numbers without VIP stimulation 25.20, with VIP stimulation 23.50, $p < 0.0001$; Wilcoxon test, Fig. S1A, exemplar recordings are shown in Fig. 2A, B) and increased inter-spike-interval (ISI) length (median SST ISI without PV stimulation 33.35 ms, with PV stimulation 39.17 ms, $p < 0.0001$; Wilcoxon test median SST ISI without VIP stimulation 39.56 ms, SST ISI with VIP stimulation 41.93 ms, $p < 0.0001$; Wilcoxon test Fig. S1B). We found a broad range of effect strengths in both connections (PV to SST: min 1.29%, max 62.42%; VIP to SST: min 0.37% max 66.53%). In two PV to SST and three VIP to SST cell pairs, we observed a massive decrease of SST cell firing (spike reduction >30%). These large effects can be explained by direct and strong hyperpolarization of postsynaptic SST cells, resulting in delayed action potential firing. We often observed inhibitory postsynaptic potentials (IPSPs) in SST cells. IPSPs were time-locked to presynaptic action potential firing and, as a result of lower firing frequencies, more easily observed in VIP to SST cell pairs (Fig. 2C, D). Our data imply that one individual cell cannot completely silence a postsynaptic neuron but is able to significantly decrease action potential firing.

Overall, spike loss through PV cell stimulation was not significantly different from that through VIP cell stimulation (PV to SST: median 11.9%; VIP to SST: median 5.9%; $p = 0.2021$; Mann-Whitney test, Fig. 2E). The same observation was also made for ISI increase (PV to SST: median 5.4 ms; VIP to SST: median 2.1%; $p = 0.3402$; Mann-Whitney test, Fig. 2F). Upon long pulse stimulation, PV cells fired significantly more action potentials than VIP cells (PV: median 166.9; VIP: median: 31.9; $p < 0.0001$; unpaired t test, Fig. 2G). When normalizing the effect strength for the number of presynaptic spikes (spike reduction divided by number of presynaptic spikes) individual PV action potentials had a significantly weaker effect on SST cell output than individual VIP action potentials (PV to SST: median 0.049%; VIP to SST: median 0.231%; $p = 0.0014$; Mann-Whitney test, Fig. 2H). Taken together, these results suggest that despite a strong difference in unitary properties of these "disinhibitory connections motifs", individual PV or VIP cells can affect their target SST cells with comparable rates of spike reduction, i.e., reduce their output.

## Spike loss correlates with various electrophysiological properties

Since PV and VIP cells are thought to differ in synapse location, and multiple studies have shown large differences in IPSC sizes[13,14,31], we did not expect an absent significant difference in spike loss between PV to SST and VIP to SST cell pairs. Therefore, we further analyzed our previous connection test recordings and explored whether we would see differential correlations in spike loss with various electrophysiological parameters: total synaptic charge transfer (total charge) of the whole train, and unitary IPSC amplitude, latency, and normalized slope. In the literature, PV inputs display much larger IPSC amplitudes than VIP inputs[13,14,31]. However, PV to SST inputs display synaptic depression, while VIP to SST inputs display facilitation at high-frequency firing[14]. Therefore, total charge of the whole train is a better measure to estimate the effect strengths of both circuit motifs during longer periods of activation[53]. Although median total charge was roughly three times larger in PV than in VIP to SST cell pairs (PV to SST: median 1.52 pC, $n = 12$, VIP to SST: median 0.56 pC, $n = 19$, Fig. 3A), they were not significantly different ($p = 0.1642$, Mann-Whitney). In both circuit motifs, total charge correlated positively with spike loss (PV to SST: $r = 0.80$, $p = 0.0025$, VIP to SST: $r = 0.52$, $p = 0.0211$, Spearman correlation, Fig. 3B).

We did not use predefined current intensities during the connectivity test and therefore observed a large variety of presynaptic action potential numbers. Thus, we also normalized the total charge for the number of presynaptic spikes. Again, we did not see significant differences between PV to SST and VIP to SST cells pairs (PV to SST: median 0.053 pC, VIP to SST: median 0.049 pC, $p = 0.9523$, Mann-Whitney test, Fig. 3C). Only in PV to SST pairs, normalized spike loss correlated positively with spike loss (PV to SST: $r = 0.88$, $p = 0.0003$, VIP to SST: $r = 0.42$, $p = 0.0687$, Spearman correlation, Fig. 3D).

Analyzing the first IPSC of the connection test, we observed cell type-specific differences between PV to SST and VIP to SST connections similar to a previous study in which we used a cesium-based intracellular solution[14]. PV to SST connections were significantly larger in IPSC amplitude (PV to SST: median 13.23 pA, $n = 14$; VIP to SST: median 3.49 pA, $n = 19$, $p < 0.0001$; Mann-Whitney test, Fig. S2A) and significantly shorter in latency (PV to SST: median 0.85 ms, $n = 15$, VIP to SST: median 2.15 ms, $n = 21$; $p < 0.0001$; Mann-Whitney test, Fig. S2B). In some cases, effect sizes were too small and signal to noise ratio was too low to adequately determine normalized slope (PV to SST: median 0.224 fraction of amplitude ms$^{-1}$, $n = 15$; VIP to SST: median 0.275 fraction of amplitude ms$^{-1}$, $n = 19$; $p = 0.5602$; Mann-Whitney test, Fig. S2C). In addition, PV to SST and VIP to SST connections display different short-term synaptic plasticity. PV to SST connections are depressing whereas VIP to SST connections were facilitating at high-frequency stimulation[14]. In the present study, we observed the same effects but did not analyze plasticity ratios in more detail due to a different scope. Two examples are shown in (Fig. 1F, G).

We calculated (Spearman) correlations of spike loss with latency and amplitude of the first response, and number of presynaptic spikes of long pulse recordings (Fig. 3E–G). For the PV to SST connection we found a strong positive correlation with amplitude ($r = 0.8374$, $p = 0.0004$) and a strong negative correlation with latency ($r = -0.7714$, $p = 0.0012$). For the VIP to SST connection we neither observed correlations with amplitude ($r = 0.0456$, $p = 0.8529$) nor latency ($r = 0.1221$, $p = 0.5981$). However, VIP to SST spike loss correlated positively with the number of presynaptic spikes ($r = 0.6545$, $p = 0.0013$), while there was no correlation with number of presynaptic action potentials for PV to SST pairs ($r = 0.2571$, $p = 0.3538$). The differences in unitary synaptic properties and correlations might hint to different mechanisms in PV and VIP cells' synaptic release. PV cells display initially strong inputs, which are then attenuated due to their synaptic depression. VIP cells, however, show facilitation at high-frequency firing which might increase the effect over time. This might also explain why we observed a positive correlation with number of spikes only in VIP cells.

## Probing the timing of postsynaptic action potential firing by a sliding window of presynaptic spike trains elicited in a presynaptic neuron

Although some cells are highly and continuously active in vivo, general whisking-evoked responses are more transient/phasic and rhythmic[54], thus long-pulse presynaptic stimulation offers a general insight on how much spike output reduction is achieved on average by the prolonged firing of a presynaptic neuron. Furthermore, long pulses lead to an averaging of action potential sequences that differ during the various time points of current injection and thus we may overlook inhibitory modulation effects that are dependent on timing that is more precise. For example, ongoing activity of PV or VIP cells may be able to delay action potential firing in postsynaptic SST cells and not only sparsify it. Therefore, we used a short-pulse protocol where PV and VIP cells were induced to fire for 100 ms. We also tested the effect of PV and VIP cell firing at different time points at beginning of the one-second-long train of evoked SST cell firing. Consequently, we used short firing pulses in presynaptic PV and VIP cells (100 ms) and tested the effect on SST cell output at multiple firing positions (pre: 50 ms before –50 ms after SST cell firing onset; early: 50–150 ms after SST cell firing onset; late: 150–250 ms after SST cell firing onset, Fig. 4A, B).

Overall, we analyzed 15 PV to SST (age: P25-P48, median age P42, from 14 animals) and 20 VIP to SST cell pairs (age: P28-P55, median age

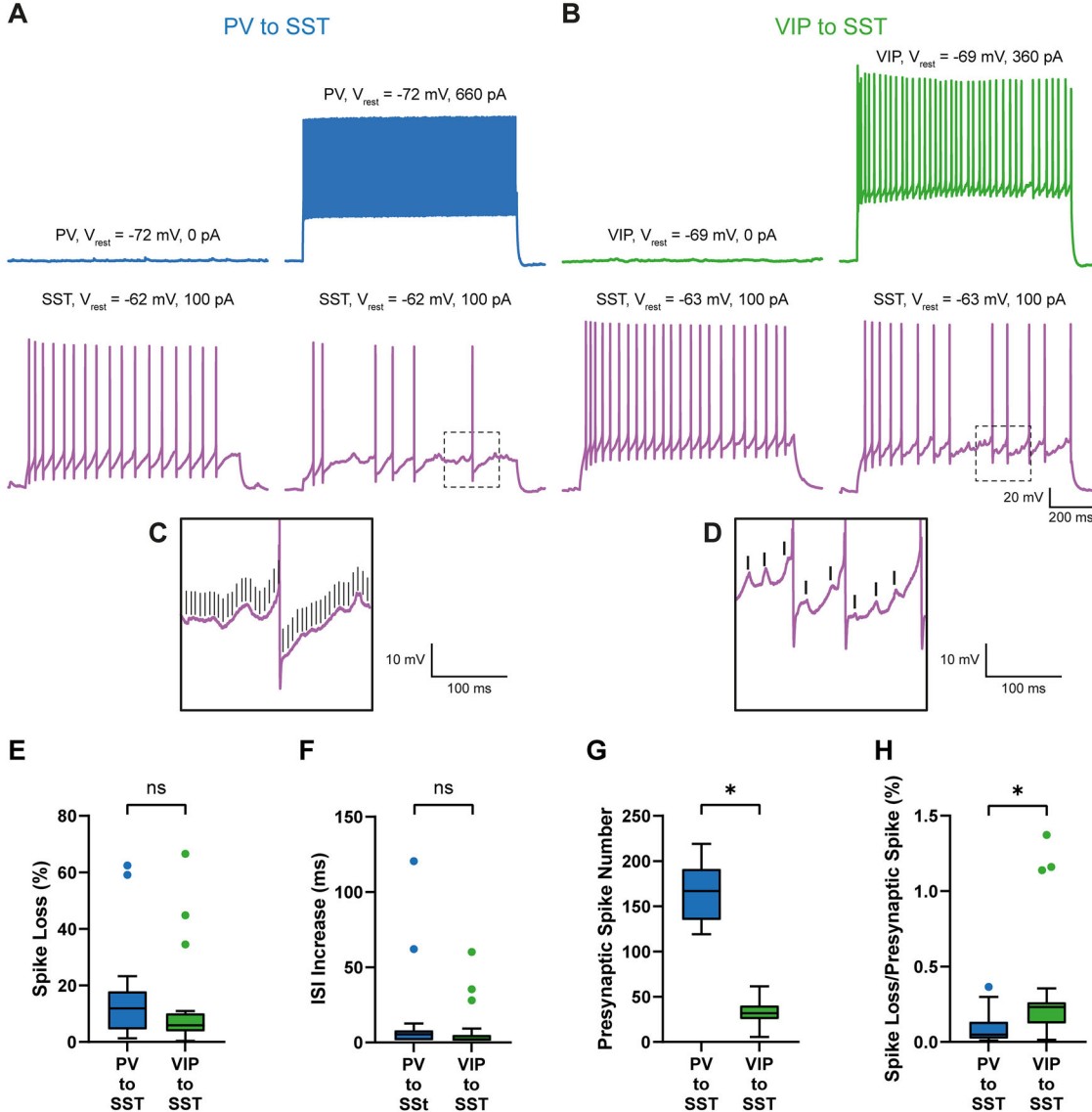

**Fig. 2 | Individual presynaptic PV and VIP cells are able to reduce the firing of postsynaptic SST cells. A** Exemplar recording of a connected PV to SST cell pair with high effect strength. Cells were recorded in current clamp mode at resting membrane potential ($V_{rest}$). Current injection solely into the SST cell leads to a robust continuous adapting firing pattern (left). Simultaneous evoked firing of a presynaptic PV and the postsynaptic SST cell resulted in a massive decrease of SST cell firing (right). **B** Same as in A but with a connected VIP to SST cell pair. **C, D** Magnification of dotted rectangles in (**A, B**). Black lines display presynaptic spike peaks of the presynaptic PV (**C**) and VIP cell (**D**). Especially in the VIP to SST cell example, time-locked postsynaptic IPSPs are clearly visible. **E–H** Quantification of the effect of PV and VIP cell firing on SST cell output (PV: $n = 15$, age range P25-

P48, median P42; VIP: $n = 21$, age P28-P55, median P35). Traces without presynaptic stimulation were always compared to the subsequent traces with presynaptic stimulation. For each cell pair we calculated the mean out of all iterations. The following parameters have been analyzed: **E** spike loss (in %) **F** ISI increase (in ms), **G** presynaptic spike number and **H** spike loss divided by the number of presynaptic spikes (in %). Within both motifs, we observed a large variability of effect strength. There was no significant difference in spike loss or inter-spike-interval increase between both groups. Fast spiking PV cells fired ≈3x more action potentials per s than VIP cells. Therefore, when normalizing spike loss to the number presynaptic spikes, effect strength of individual VIP spikes was significantly larger than of PV cells. Asterisks indicate significant differences between groups ($p < 0.05$).

P35, from 18 animals). Since the stimulation at the first time point started 50 ms before SST cell current injection, we could show that both cell types were able to significantly delay the first spike (PV to SST: median first SST spike after current injection shifts from 13.97 to 14.40 ms, $p = 0.0009$, Wilcoxon test; VIP to SST median first SST spike after current injection shifts from 15.16 to 15.86 ms, $p = 0.0309$, paired t test Fig. S3A, B). When comparing both connection types, we did not observe a significant difference between PV to SST and VIP to SST cell pairs (PV to SST: median delay of first SST spike 0.586 ms; VIP to SST: median delay of first SST spike 0.677 ms; $p = 0.5644$, Mann-Whitney test, Fig. 4C). For quantification of spike loss, ISI increase, and normalized spike loss (spike loss divided by number of presynaptic spikes), we always compared PV to SST with VIP to

SST cell pairs of the same time point of induced presynaptic firing, respectively. In summary, in none of the tested parameters we found significant differences between PV to SST and VIP to SST cell pairs at the respective firing position (Fig. 4C–F). Data including p-values can be found in Table S1. Since we used a much shorter pulse (100 ms), loss of individual action potentials had a stronger impact on SST cell firing in comparison to the long pulse. Therefore, we observed larger maximal spike loss for PV to SST (87.5%) and VIP to SST cell pairs (80%).

### Putative contact site analysis
Effect strength of inhibitory inputs of various IN on spike generation in postsynaptic neurons might be influenced by synapse location since distal

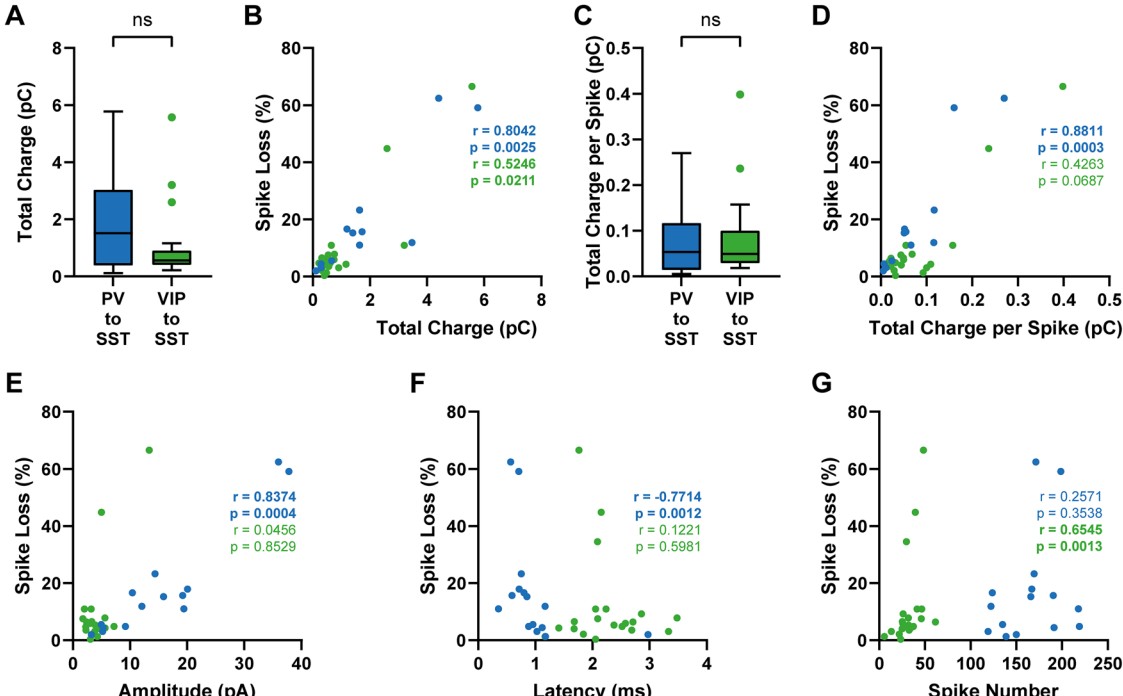

**Fig. 3 | Correlation of various electrophysiological parameters with effect strength displays cell type specific differences but total synaptic charge does not.** **A–D** Quantitative analysis and Spearman correlation of total charge and normalized total charge for the number of presynaptic spikes. **A** Despite median total charge in PV to SST cells being nearly three times higher, there was no significant difference between PV to SST and VIP to SST cell pairs ($p = 0.1642$). **B** Total charge of both circuit motifs correlated positively with effect strength. **C** Again we did not see significant differences between both motifs. **D** Spearman correlation of normalized total charge with spike loss only revealed a significant positive correlation in the PV to SST circuit motif. PV to SST: $n = 12$, VIP to SST: $n = 19$. Spearman correlation of spike loss with amplitude (**E**), latency (**F**), and presynaptic spike number (**G**). PV to

SST pairs are shown in blue and VIP to SST pairs in green. PV to SST: $n = 15$, VIP to SST: $n = 21$ (**F**, **G**). Connection tests for one PV to SST and two VIP to SST cell pairs were recorded in current clamp. Therefore, we were not able to calculate amplitudes in these cases (**E**). In PV to SST pairs, we observed a negative correlation with amplitude and spike loss, a positive correlation with amplitude and spike loss but no correlation with the number of presynaptic spikes and spike loss. In VIP to SST pairs, there was no correlation with the number of presynaptic spikes and spike loss. However, we observed a positive correlation with the number of VIP spikes and spike loss. The correlation coefficient is given as r. Correlations with $p < 0.05$ were interpreted as significant.

dendritic inputs are attenuated and delayed[55]. Therefore, we determined the number and location of putative contact sites (PCS) in morphological reconstructions (exemplar reconstructions are shown in Fig. 5A, B, all reconstructed cell pairs are shown in Figs. S4 and S5). Recorded cell pairs were visualized using Alexa Fluor 633-conjugated streptavidin. We reconstructed the four best-recovered cell pairs with various effect strengths of PV to SST and VIP to SST cell connections, respectively, and performed PCS analysis (median spike loss of PV to SST: 41.20%; median spike loss VIP to SST 27.89%; $p = 0.6611$; unpaired t test, Fig. S6A). PCS were identified during reconstruction on image stacks of a large cortical volume and afterwards re-imaged using enhanced confocal microscopy to further validate PCS at close-to-super resolutio[56]. Exemplar PCS are shown in Fig. 5A, B). We analyzed the total number of PCS, distance from soma along the dendritic tree, and in case of dendrite-originating axons whether PCS were located on this specific dendritic branch. In all pairs, except one, high-resolution confocal microscopy decreased the number of PCS estimated by standard confocal microscopy (PV to SST: from 6, 4, 9, 9 PCS to 3, 2, 4, 2 PCS; VIP to SST: from 17, 4, 11, 4 PCS to 3, 4, 3, 1 PCS; Fig. S6B). Overall mean distance of PCS was not significantly different in both connections (median PV to SST 87.55 μm; median VIP to SST: 112.4 μm; $p = 0.1194$; unpaired t test, Fig. 5C, Fig. S6C). Number of PCS was not correlated with spike loss (PV to SST: Pearson $r = -0.1946$, $p = 0.8054$; VIP to SST: Pearson $r = 0.5946$, $p = 0.4055$; Fig. S6D). In all but one SST cells, the axon originated from a proximal dendrite. In two PV to SST and two VIP to SST cell pairs we found PCS on this very dendrite. In individual PV to SST and VIP to SST cell pairs we found PCS in the perisomatic region (<50 μm from soma). However, mean PCS distance did not significantly correlate with spike loss (PV to

SST: Pearson $r = 0.0712$, $p = 0.9288$; VIP to SST: Pearson $r = -0.5686$, $p = 0.4314$; Fig. 5D). We also tested whether spike loss correlated with distance of the nearest PCS. Again, we did not see a significant correlation (PV to SST: Pearson $r = -0.2956$, $p = 0.7044$; VIP to SST: Pearson $r = -0.55$, $p = 0.4488$; Fig. 5E). Thus, no obvious structure-function correlate could be detected at the level of high-resolution light microscopical analysis.

## Discussion

A major question in neuroscience is whether a single neuron can exert a meaningful effect on its target cells or whether an ensemble of presynaptic neurons is necessary to change the output of a postsynaptic cell[57]. Here we have probed this question for so called disinhibitory circuit motifs, i.e., connections of inhibitory neurons onto other inhibitory neurons[13]. The present study suggests that PV to SST and VIP to SST cell connections are both capable to reduce the output of postsynaptic SST cells in layer 2/3 of barrel cortex although they substantially differ in unitary synaptic properties and short-term plasticity[14,17,31]. Paired recordings revealed that action potential induction in individual presynaptic PV or VIP cells led to a significant decrease in firing of postsynaptic SST cells that were identified as Martinotti cells. Given the finding that putative contact sites of the studied connections did not differ in terms of number and location, our results question the current notion that PV basket cells are exclusive output controllers due to perisomatic targeting of postsynaptic neurons and suggest VIP cells, beyond modulating dendritic integration, may act as an alternative type of output controller.

Here, we used somatic recordings in acute brain slices. By direct somatic current injection in both neurons, we could titrate the number of

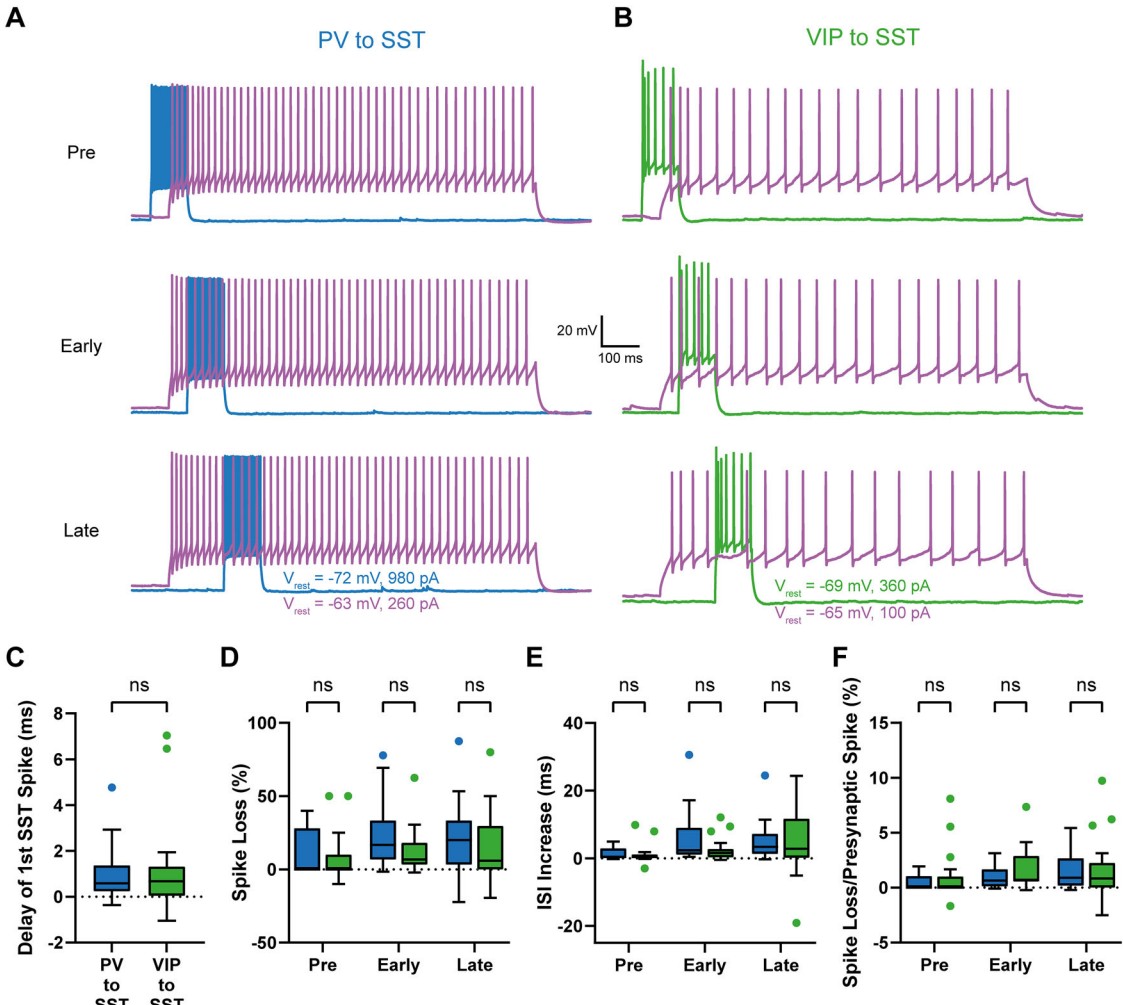

**Fig. 4 | A sliding window of brief presynaptic firing of PV and VIP cells is equally effective to delay/suppress firing of postsynaptic SST cells.** Exemplar short pulse recordings of a PV to SST (**A**) and a VIP to SST (**B**) cell pair. We analyzed three different time slots in respect to postsynaptic SST cell firing times (pre: 50 ms before until 50 ms after postsynaptic firing onset, early: 50–150 ms after postsynaptic firing onset, late: 150–250 ms after postsynaptic firing onset), in order to test whether SST cell firing is more vulnerable to presynaptic activity at various time points. SST cell firing is shown in magenta, PV in blue and VIP in green. Note that presynaptic firing reproducibly led to reduced SST cell firing in the respective time window. In the pre window, presynaptic activity resulted in a visible sag in SST cell membrane potential.

This resulted in a delay of the first SST cell action potential (**C**) which was again not significantly different between both motifs. Quantitative analysis of spike loss (**D**), ISI increase (**E**), and spike loss per number of presynaptic spikes (**F**) at time slots pre, early, and late. Overall, we did not find any significant differences between the PV to SST and VIP to SST connection in any analyzed parameter at any time slot. Due to the smaller analysis window, (100 ms) depletion of individual spikes had a stronger effect than in the long pulse data set. Overall, we observed higher effect strengths (max spike loss: 87.5%). In some pairs, we saw increased spike numbers, which we did not see previously. PV to SST: $n = 15$, age: P25-P48, median P42; VIP to SST: $n = 20$, age P28-P55; median P35. Dotted lines in **D**–**F** indicate the zero line.

evoked spikes in the presynaptic neuron to reliably and reproducibly evoke action potential firing independent from integration of excitatory inputs on the dendritic tree. Current intensities were chosen to ensure action potential firing throughout the full one second sweep and reducing sweep-to-sweep differences for better analysis. Overall, firing frequencies are in range of what can be observed in vivo[54]. There, interneurons display phases of spontaneous activity for multiple seconds during quiet wakefulness and active whisking[54,58,59]. Therefore, we used one-second-long pulses to test the maximum effect strength of presynaptic activity, which resembles the continuous activity in vivo. Moreover, VIP and PV inputs to SST cells display different unitary synaptic properties and short-term plasticities. PV to SST inputs are initially large in amplitude and depressing, whereas VIP to SST inputs are initially small but facilitating at high frequency stimulation[14]. These differences were most prominent in the early phases of action potential firing (see Fig. 1F, G). We assumed that these opposing effects level out during prolonged stimulation, which would mask cell type-specific differences. Already

200 ms-long current pulses were too long to see significant differences in total charge whereas 50 ms would have been too short to generate a sufficient amount of action potentials. Therefore, we used a 100 ms-long pulse and tested this at different phases of SST cell firing.

We consider that a mixture of two biophysical mechanisms can explain the observed effect on action potential firing in SST cells. On the one hand, opening of GABA$_A$ receptor-coupled chloride channels leads to a conductance change at input sites, which results in less effective current and lower overall input resistance, thereby reducing the effectiveness of somatic current application. On the other hand, current application depolarizes the cells, which increases the driving force for chloride ions and thereby enhances GABA-mediated inhibitory currents hyperpolarizing the cell. Indeed, we were able to see IPSPs in our paired recordings (see Fig. 2C, D). However, currently there is no methodological approach how to disentangle both effects. Dendritic recordings might be a useful technique to analyze local conductance changes, although dendritic recordings in interneurons are very challenging[60,61].

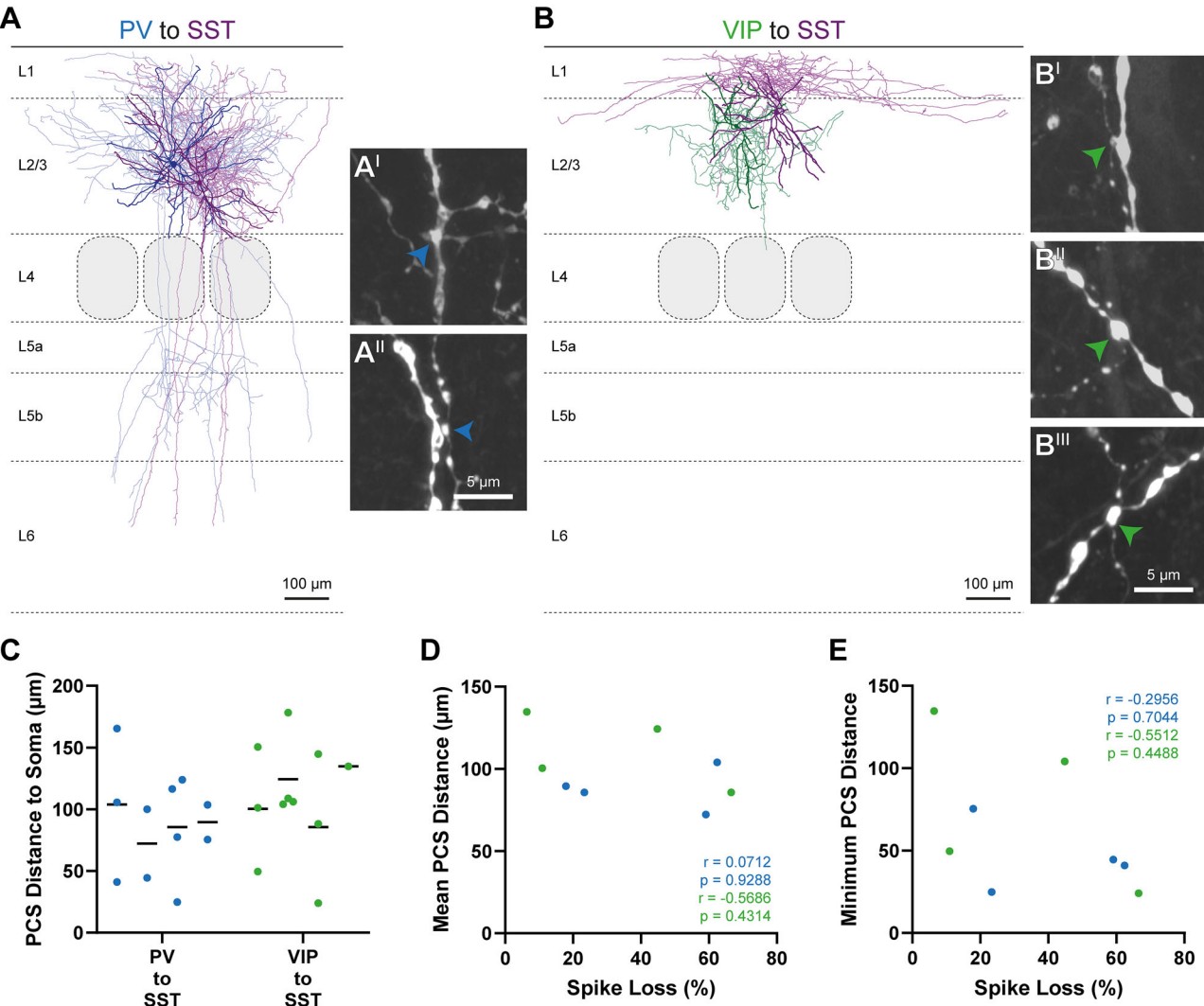

**Fig. 5 | Putative contact site analysis of morphologically reconstructed PV to SST and VIP to SST pairs.** We reconstructed four PV to SST and VIP to SST cell pairs, respectively, displaying various effect strengths and performed PCS analysis. Morphological reconstruction of a PV to SST (**A**) and a VIP to SST (**B**) cell pair with associated PCS images. Soma and dendrites are shown in dark purple and axon in light purple. Both SST axons show intense innervation of L1 and can therefore be characterized as MC. PV soma and dendrites in dark blue and axon in light blue. VIP soma and dendrites are shown in dark green and axon in light green. The PV cell can be classified as BC and the VIP cell as multipolar cell. In the PV to SST cell pair, we found two PCS ($A^I$, $A^{II}$) and in the VIP to SST cell pair three ($B^I$, $B^{II}$, $B^{III}$). PCS images are shown as partial Z-projections to better visualize axonal trajectory. PCS were counted when axonal boutons got into gapless contact with dendritic structures, by coursing either in close proximity or by directly crossing it. PV PCS are shown with blue and VIP PCS with green arrowheads. Scale bars for the reconstruction are 100 μm, and for the PCS, 5 μm. **C** Number and distance of PCS from the soma along the dendrite from all reconstructed cell pairs. Individual pairs are represented in a column. Lines mark the mean distance. Although PV cells are thought to be soma-targeting, we did not find any somatic PCS in any reconstructed cell pair. **D** Spearman correlation of mean PCS distance to soma and spike loss. PV to SST cells pairs are shown in blue and VIP to SST cell pairs are shown in green. **E** The same as in D but plotted with the shortest PCS distance to soma. Overall, there was no significant correlation with PCS distances and spike loss in both analyzed circuit motifs. The correlation coefficient is given as *r*. Correlations with *p* < 0.05 were interpreted as significant.

In addition to confirming their differences in unitary synaptic properties, our experiments revealed a previously undocumented feature: individual PV and VIP cells are able to significantly decrease action potential firing of postsynaptic SST cells. Surprisingly, when comparing the PV to SST and VIP to SST connection there was no significant difference in spike loss and ISI increase. However, PV cells fired substantially more action potentials than VIP cells per stimulation epoch. Therefore, we additionally compared spike reduction after normalization for the number of presynaptic action potentials. This way, the effect of individual VIP spikes on SST cell spike output was significantly larger than of individual PV spikes. Effect strength was only correlated with amplitude and latency in PV to SST and not in VIP to SST connections. In contrast, number of presynaptic action potentials was correlated with spike loss in VIP to SST and not in PV to SST cell pairs. The more action potentials they fired, the stronger the VIP

cell-induced effect was. This might also explain why we did not see significant differences in normalized effect strength of short pulse experiments as VIP cells fired less action potentials there.

We suggest that this effect results from the differences in short-term synaptic plasticity. The PV inputs are strong at stimulus onset but rapidly depress, decreasing their effect on postsynaptic target cells. Depression is a general feature of PV synapses and has been described in multiple studies independent form postsynaptic target cells[14,31,62]. The VIP effect might result from the interplay of different mechanisms. Synaptic facilitation at high-frequency stimulation, which has already been described in L2/3 VIP to SST connections in mouse barrel cortex[14] and temporal summation of inhibitory currents further amplifying the facilitating effect. However, the normalized effect was only significantly different in the long-pulse experiment. We did not observe significant differences using a short presynaptic pulse of action

potential generation. In addition, synaptic plasticity of VIP cells reported in the literature is not as uniform as for PV cells. We reported facilitation at high frequency stimulation in Walker et al.[14], other studies describe no effect on plasticity[31] or even depression[16,63]. These discrepancies in VIP short-term plasticity might be explained by differences between cortical areas as studies were performed in visual cortex[31], motor cortex[63], and auditory cortex or prefrontal cortex[16]. In our recent study, we also observed differences in short-term plasticity between barrel cortex and visual cortex in a translaminar L2/3 VIP to L4 SST circuit[64]. Therefore, it remains to be determined whether VIP cells would display the same effect strengths on spike output reduction in SST target cells outside L2/3 of primary somatosensory cortex or in other cortical areas.

Our findings might sound counterintuitive in that two cells types, which are fundamentally different in unitary synaptic properties and short-term plasticity, have virtually the same effect on the output of SST cells. Nevertheless, VIP and PV cells can be recruited via different pathways. PV cells are activated via local PN and thereby provide feedforward and feedback inhibition on the local microcircuit[1,29,65]. Integration of VIP cells is more diverse and involves a multitude of local and long-range inputs[15,33,66,67]. Thereby, different brain regions may disinhibit PN via distinct pathways.

In a recent study in mouse hippocampus, Chamberland, et al.[38] analyzed the effect of individual GABAergic IN firing on postsynaptic PV cells. They observed that burst and even single spike IN firing can interrupt PV cell activity for multiple hundreds of milliseconds in vitro and in vivo. Chamberland, et al.[38] suggest a cell-autonomous mechanism involving D-type potassium currents, which are partly mediated by the voltage-gated potassium channel 1.1 ($K_V 1.1$). In our study, we did not analyze SST to PV connections in detail. However, 84.2% (16/19) of PV to SST pairs were reciprocally connected and we did not observe any persistent interruption of firing.

PV cells are thought to be perisomatic inhibitors that target the soma and proximal dendrites of their target cells[41,68], while VIP cells are thought to predominantly target dendrites[44,69]. Nevertheless, there are studies that displayed a strong innervation of PV cell somas by VIP cells[70,71]. However, this specificity of VIP synapses is not observed for all cell types, since the overall amount of soma-targeting VIP synapses remains below 10% in L2/3 of the rodent neocortex[44].

We observed a large individual variation of effect strength within PV or VIP to SST cell connections but no overall significant difference. Therefore, we performed putative contact site (PCS) analysis to test whether number or location of synapses on the dendritic tree of target cells might explain this effect. In both circuit motifs, we analyzed cell pairs with weak and strong effect sizes. Despite their assumed difference in synapse location, PCS distance from soma was not significantly different between reconstructed PV to SST and VIP to SST cell pairs. Surprisingly, we did not find somatic PCS in PV to SST cell pairs and only three of eleven PV PCS were located on perisomatic dendrites (distance <50 µm from soma). In addition, neither mean PCS distance nor distance of closest PCS correlated significantly with spike loss. One has to mention that our PCS analysis was based on light microscopy, which might lead to inaccurate PCS numbers[72–74]. Nevertheless, in a recent study, Schneider-Mizell, et al.[75] used electron microscopy to analyze the connectome of mouse visual cortex. They found numerous PV cells that target the soma of excitatory neurons. Inhibitory synapses, however, often are also more distally located on the dendritic tree. Schneider-Mizell, et al.[75] classified cells due to their synaptic targeting. Nevertheless, the strong somatic targeting of excitatory cells in combination with morphology might be a good indicator to correctly identify PV cells. Our data suggests that PV cell properties like soma-targeting should be tested for various target cells and not be generalized from previous knowledge obtained with pyramidal cells only.

The PV interneuron subpopulation can be considered the "allrounder" and the primary workhorse in all existing forms of inhibition, including disinhibition[2,4,21]. During bouts of foveal whisking, in order to achieve decision making in a tactile discrimination paradigm, the PV cell-mediated inhibition/disinhibition circuit motifs may be tuned to achieve fast evidence accumulation and consequently correct decision making[76] due to the high inhibitory current flow right from the start of their firing. By contrast, disinhibition more than inhibition may dominate VIP cell-mediated effects and, due to initially weak but then increasing inhibitory current flow, with some delay, they may open input channels[77] that are necessary for more refined and extended evidence accumulation during longer bouts of foveal whisking before a decision can be made.

Our thinking on disinhibition has to be refined in the sense that every GABAergic interneuron subpopulation (also the ones not mentioned here, like somatostatin-expressing neurons or LAMP5/NDNF-expressing neurogliaform cells) seems to be involved -to a certain extent- in disinhibitory circuit motifs[78,79]. So the question in the future should not be "Can a certain type of IN perform disinhibition?" but when and how does it do it.

In summary, we were able to show that individual L2/3 PV and VIP cells are actually capable to impose output control on L2/3 SST cells in mouse barrel cortex. PV and VIP cell stimulation led to decreased output of SST cells with various effect strengths within both connection types but proving them to be effective inhibitors already at a single cell level. Our findings furthermore challenge the general concept of a strict division of labor of IN subtypes attributing output control solely to perisomatic inhibitors and input control to dendritic inhibitors in mouse barrel cortex.

## Methods

### Animals

All experiments were performed in accordance to the German guidelines of animal care. The experimental protocol was approved by the animal welfare officer of the University Medical Center Göttingen. Mouse lines were bought from the Jackson Laboratory (Bar Harbor, ME, USA). All mice were housed in standard housing conditions with access to food and water *ad libitum* and a 12 h light/dark cycle. PV-Cre (B6;129P2-Pvalbtm1(cre)Arbr/J, strain #008069) or VIP-Cre (VIPtm1(cre)Zjh/J, strain #010908) mice were crossbred with GIN mice (FVB-Tg(GadGFP)45704Swn/J, #003718) to generate PV-Cre//GIN and VIP-Cre//GIN mice. Double transgenic mice were further crossbred with Ai9 mice (B6.Cg-Gt(ROSA)26Sortm9(CAG-tdTomato)Hze/J, strain #007909) to generate PV-Cre//tdTomato//GIN and VIP-Cre//tdTomato//GIN mice. GIN stands for GFP-expressing inhibitory interneurons and in this mouse line a subpopulation of SST cells express EGFP[50]. PV and VIP cells can be identified by their tdTomato expression. Since 100% of GIN cells also express SST[51] we are referring to GIN cells as SST cells.

### Brain slice preparation

Mice of both sexes (postnatal days 24-55, median: 35) were deeply anesthetized with isoflurane and decapitated. Brains were quickly removed and transferred into constantly oxygenated (Carbogen: 95% $O_2$/5% $CO_2$) ice cold cutting solution (in mM: 87 NaCl, 1.25 $NaH_2PO_4$, 2.5 KCl, 10 glucose, 75 sucrose, 0.5 $CaCl_2$, 7 $MgCl_2$ and 26 $NaHCO_3$; pH 7.4). Hemispheres were separated and 300 µm thalamocortical slices were cut according to Porter, et al.[80]. Slices containing the barrel field were transferred to oxygenated artificial cerebral and spinal fluid (ACSF, in mM: 125 NaCl, 1.25 $NaH_2PO_4$, 2.5 KCl, 25 glucose, 2 $CaCl_2$, 1 $MgCl_2$, 26 $NaHCO_3$) and incubated at 32 °C for 30–45 min. After incubation, slices were kept at room temperature until further use.

### Electrophysiology

Brain slices were transferred to a recording chamber in an upright microscope (Axio Examiner, Zeiss, Oberkochen, Germany) with an ACSF flow rate of ca. 2 ml s$^{-1}$ at 32 °C. Whole cell patch clamp recordings of GIN, PV and VIP cells were performed in current and voltage clamp mode. Freshly pulled (P-1000 Micropipette Puller, Sutter Instruments, Novato, CA, USA) borosilicate glass pipettes (Science Products, Hofheim, Germany) with a resistance of 5–8 MΩ were filled with a potassium gluconate-based intracellular solution (in mM: 135 K-gluconate, 5 KCl, 0.5 EGTA, 10 HEPES, 4 Mg-ATP, 0.3 Na-GTP, 10 Na-phosphocreatine-phosphate, pH 7.4). Before experiments, 0.3–0.5% biocytin was added to the pipette solution to allow

staining of patched cells after experiments. Passive and active cell properties were characterized using hyper- and depolarizing 1 s current pulses (starting at −100 pA with 10 pA increments). Data were acquired using SEC-05 X amplifiers (npi electronic, Tamm, Germany) in discontinuous mode with a switching frequency of ≈50 kHz, filtered at 3 kHz and digitized at 10–25 kHz using a CED Power 1401 interface (CED Limited, Cambridge, England).

## Paired recordings

To keep the connection probability high, we mostly recorded from cell pairs in close proximity to each other (<200 µm). In all our experiments, GIN cells were the designated postsynaptic cells. Connections were tested by injecting a 200 ms current pulse into putative presynaptic PV and VIP cells to induce high-frequency firing. GIN cells were held in voltage clamp close to firing threshold (ca. −45 mV). In some cases, GIN cells were recorded in current clamp mode. Cells were depolarized to increase the driving force for GABA-mediated currents to facilitate detection. A short test pulse (200 ms, −5 mV) was used in postsynaptic SST cells to analyze the access resistance ($R_a$). $R_a$ in SST cells was 10.03 ± 0.34 MΩ (mean ± s.e.m.; $n = 40$). Recordings which were incomplete or did not meet long-term recording quality standards, were not further analyzed. Nevertheless, all pairs in which we were able to clearly demonstrate a connection were included in these connectivity rate measures (Fig. 1F, G). We defined neuron pairs as connected when we observed multiple time-locked IPSCs with latencies (time from presynaptic spike peak to IPSCs onset) below 3 ms during the 200 ms high frequency firing connection test.

To analyze the effect of presynaptic PV and VIP cell firing on post-synaptic SST cells, we used one-second-long current pulses to generate action potentials. For PV and GIN cells, the current injection strength was set to double rheobase and for VIP cells to triple rheobase. At these current injection strengths, all cell types displayed cell type-specific firing patterns throughout the whole stimulus duration. Rheobase was determined by using one-second-long current injection pulses with a step size of 5 pA. The lowest current which resulted in action potential firing was defined as rheobase. All recordings were performed at resting membrane potential $V_{rest}$ (PV: ca. −62 mV, VIP: ca. −69 mV, SST: ca. −62 mV). Now, we first recorded GIN cell firing without concurrently induced presynaptic firing. After a delay of 10 s in between sweeps, we subsequently recorded GIN cell firing with simultaneously evoked presynaptic PV or VIP cell firing. In total, we recorded 10 iterations without presynaptic and 10 iterations with pre-synaptic stimulation. As a complementary approach, we also used a short 100 ms-long shifting presynaptic square pulse (short-pulse) with the same current strength at three different time point with respect to postsynaptic firing: (i) a first starting 50 ms before the postsynaptic pulse (a stimulus window called "pre"). After recordings of three iterations with and without presynaptic stimulation, the short pulse was moved to (ii) a second position (called "early": 50–150 ms after postsynaptic firing onset) and finally to (iii) a third position (called "late": 150–250 ms after postsynaptic firing onset) to test for potential differential susceptibility of the postsynaptic neuron output to time-restricted inhibitory input.

The effect on postsynaptic SST cell firing was analyzed by comparing the sweep without presynaptic stimulation to the following sweep with simultaneous presynaptic stimulation and was averaged for each pair, respectively. The following parameters were analyzed: (i) spike loss (post-synaptic spike reduction through presynaptic stimulation, in %), (ii) interspike-interval-increase (increase of time between postsynaptic action potentials, in ms), (iii) presynaptic spike number (number of presynaptic action potentials) and (iv) normalized spike loss (spike loss divided by number of presynaptic action potentials). For the shifting pulse, we also analyzed the delay of the first action potential at the first presynaptic time slot ("pre").

Unitary synaptic properties were analyzed as in Walker, et al.[14], namely postsynaptic responses of the first action potential of the connection test were aligned with respect to spike peak and averaged. We analyzed (i) IPSC amplitude (difference from baseline to peak), (ii) latency (time between presynaptic spike peak and IPSC onset (>3x baseline standard deviation)), and (iii) normalized slope of the ascending phase (average slope, determined by means of a least-square best fit, divided by the amplitude). Furthermore, synaptic charge transfer (total charge) of each connection was calculated as the integral of full train IPSCs averages. Since we did not aim for comparable action potential numbers during our connection test initially, we also normalize the total charge for the number of presynaptic action potentials.

## Immunohistochemistry

After recordings, recording electrodes were carefully retracted to ensure re-closure of cell membranes and minimize biocytin-evoked background noise in images. Slices were fixed in 4% paraformaldehyde solution with 15% picric acid at 4 °C for at least 24 h. Slices were washed with phosphate buffered saline (PB, 0.1 M, pH 7.4) until picric acid was removed, following two more washing steps with PBS for 15 min. Slices were then washed in TRIS buffer (TB; 2 × 15 min; 0.05 M, pH 7.6), TRIS-buffered saline (TBS; 2 × 15 min), TBS + 0.5% Triton-X 100 (TBST; 2 × 15 min). Unspecific binding sites were blocked using a bovine serum albumin (BSA) solution (0.25% BSA, 10% normal donkey serum and 0.5% Triton X-100, pH 7.6 in PBS) for 90 min. Primary antibodies (goat α-GFP, 1:2000, Abcam, Cambridge, UK and rabbit α-RFP, 1:500, Rockland, Limerick.PA, USA) were diluted in BSA solution and incubated for 48–72 h at 4 °C. Slices were washed with TBST buffer (4×15 min) and incubated with secondary antibodies (donkey α-goat AF488 and donkey α-rabbit AF546 (both 1:500, Molecular Probes, USA)), and streptavidin-conjugated AF633 (1:300)) in TBST buffer. After three washing steps with TBST (15 min) and one with TBS buffer (15 min), slices were incubated with 4',6-diamidino-2-phenylindole (DAPI, 1:5000) in TBS buffer. Finally, slices were washed with TBS (1 × 15 min) and TB buffer (2 × 15 min) before they were mounted in AquaPolyMount solution (Polysciences, Warrington, PA, USA) and covered with a coverslip (24 × 5 mm, 0.08–0.12 mm thickness, Menzel).

## Morphological reconstruction and putative contact analysis

Overview images (10x magnification) of patched cells were taken using an upright microscope (Axio Observer, Zeiss) using Zen Blue software (Zeiss). If both cells were reasonably recovered, we took images of the whole cortical area containing biocytin-filled processes using a confocal microscope (Zeiss LSM 880). We used different water and oil objectives (40x and 63x) and as imaging technique either classical confocal microscopy or Zeiss Airyscan fast mode[56]. For stitching, we either used ZEN Black software (Zeiss) or the Grid/Collection Stitching plugins in Fiji[81]. Image stacks were downsized (10x) and neurons were reconstructed using Neurolucida software (MBF Bioscience, Colchester, USA). Dendrites were distinguished from axons by their diameter, fine structure and branching patterns. After reconstruction, we performed PCS analysis. PCS were validated using enhanced confocal microscopy (Zeiss Airyscan[56]) using a 63x oil objective and a step size of either 0.1 or 0.2 µm to further validate or rule out putative contact sites. PCS were counted when axonal boutons got into gapless contact with dendritic structures, either by coursing in close proximity (in the XY-plane) or by directly crossing it (without a gap in the Z-axis).

## Statistics

Statistical analysis was performed using GraphPad Prism (GraphPad Software, Boston, MA, USA). All data was tested for normal distribution using the Shapiro Wilk-test. For unpaired data that was normally distributed, an unpaired two-tailed Student t-test and otherwise a Mann-Whitney U test was used. For paired data that was normally distributed, a paired student t-test was used, for not normally distributed data, the Wilcoxon matched-pairs signed rank test was used. If correlation data was normally distributed, correlation was computed using the Pearson correlation analysis, otherwise the Spearman correlation analysis was used. Data are described as median (if not stated differently) and visualized as Tukey box and whisker plots. $p < 0.05$ were interpreted as significantly different.

## Reporting summary

Further information on research design is available in the Nature Portfolio Reporting Summary linked to this article.

## Data availability

Numerical source data underlying the graphs are provided as Numerical_Source_Data in the Supplemental data files. Any additional raw data can be obtained from the corresponding author upon reasonable request. The original data are available upon request form the corresponding author. See Reporting summary for details

## Code availability

The original analysis code written in Signal 5 can be obtained on Zenodo. (https://doi.org/10.5281/zenodo.19664110)

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

## Acknowledgements

We thank Patricia Sprysch, Sandra Heinzl and Pavel Truschow for their excellent technical assistance and Leander Matthes, Paul Molis, Sabrina Hübner, Luca Auge and Christin Korb for morphological reconstructions. This study was supported by grants from the Deutsche Forschungsgemeinschaft (DFG, STA 431/14-1; 21-1).

## Author contributions

F.P. carried out experiments, F.P., M.M., M.W. and J.F.S. were involved in study design. F.P. carried out data analysis. F.P. and J.F.S. wrote the paper. J.F.S. supervised all experiments and secured funding.

## Funding

## Competing interests

The authors declare no competing interests.
