## [Transparent Peer Review file · Communications Biology]

Cortical PV and VIP interneurons similarly influence SST neuron output despite distinct unitary properties

Corresponding Author: Professor Jochen Staiger

Version 0:

Reviewer comments:

Reviewer #1

(Remarks to the Author)

The manuscript entitled “Consequences of individual PV and VIP interneuron firing on the output of postsynaptic SST neurons in mouse barrel cortex” by Preuss et al. investigates the innervation and modulatory capacity of PV and/or VIP interneurons on action potential firing in postsynaptic SST interneurons. The authors approach these questions by combining dual recordings between interneuron pairs with morphological reconstructions of recorded pairs. Overall, the manuscript is well written and the introduction offers a good overview of the current literature regarding interneuronal output control. The study finds high coupling probabilities between pairs of PV-SST and VIP-SST interneurons. In a lot of cases, these connections appear reciprocal. In contrast to the notion that PV interneurons provide somatic inhibition whereas VIP interneurons tend to provide dendritic inhibition, Preuss et al find that both interneuron types exert a comparable output control in SST interneurons. The discussion recapitulates the findings of the present study and integrates these findings into current concepts of interneuronal output control.

While the study offers new details on disinhibitory circuit motifs between PV/VIP interneurons and SST interneurons, there are some (mainly methodological) concerns that need to be addressed:

- What is the estimated chloride equilibrium potential in the authors’ experimental settings? It is surprising to see somatic (“electrode-seen”) uIPSPs in Fig. 2D at an estimated SST interneuron membrane voltage of -40 mV or less?
- What were the exact parameters according to which a synaptic connection was determined (a) successful or (b) unsuccessful?
- Overall, the authors report very high coupling rates between pairs of recorded interneurons. Can the authors discuss/speculate on the reasons of finding such high coupling rates in contrast to other studies (e.g. Jiang et al., 2015, PMID:26612957; Campangola et al., 2022, doi: 10.1126/science.abj5861)?
- Could the authors provide exemplar recordings of reciprocally connected cell pairs?
- Given that the authors found +80% of reciprocally connected PV-SST interneuron pairs, it would have been interesting to compare SST interneuron output to either PV and VIP interneuron output. Has this analysis been done in some cells?
- Have the authors considered inherent sweep-to-sweep variability of SST interneurons upon identical current pulse injections in their analysis?
- Could the authors please provide an explanation of why uIPSC amplitudes/latencies are partly divergent from their previous study (Walker et al., 2016, <https://doi.org/10.1038/ncomms13664>)?
- The authors might want to include a quantitative comparison of phase planes of synaptically delayed action potentials in SST interneurons.
- The authors should consider a preliminary assessment of postsynaptic GABAergic receptors on SST interneurons as another powerful output modulator.

Reviewer #2

(Remarks to the Author)

While the manuscript addresses inhibitory interactions among interneuron subtypes, the novelty of this observation is somewhat limited, as it is already well established that interneurons can inhibit one another. The strength of the work lies in the effort to dissect specific inhibitory motifs between PV, VIP, and SST neurons; however, unresolved methodological and interpretational concerns substantially limit the overall impact and contribution of the study. Importantly, this study builds

directly on the authors' earlier work (Walker et al., 2016) and, while it provides additional detail, it reads more as a continuation of that study rather than a substantial advance beyond it.

Lines 83–86: The authors state that VIP cells primarily target dendrites. However, previous studies have shown that VIP neurons can also innervate the soma of other interneurons (e.g., *Eur J Neurosci.* 2007 Apr;25(8):2329–2340. doi: 10.1111/j.1460-9568.2007.05496.x).

Line 113, Materials and Methods section: The authors state that experiments were performed on mice of both sexes, postnatal days 24–44 (median: P34). This is inconsistent with other parts of the manuscript. For example, on line 279, it is reported that recordings and analyses of 15 PV → SST pairs were performed in mice aged P25–P48 (median: P42, from 14 animals).

Line 209: Morphological reconstruction and putative contact analysis.

Although Airyscan imaging improves upon standard confocal resolution, the method as described is not sufficient to confirm synaptic connectivity. Identifying putative contact sites based solely on gapless apposition risks a high false-positive rate, as axons and dendrites frequently pass in proximity without forming synapses. The lack of systematic validation with synaptic markers or electron microscopy makes it difficult to assess the accuracy of the reported PCS. Consequently, the conclusions drawn from these counts should be interpreted with caution until supported by orthogonal approaches.

Line 242–244: In a previous study, we were able to show that in triple transgenic animals, GIN cells in L2/3 can all be classified as MC, which are targeted by PV and VIP cells. The reference is missing.

Line 249–251: The author's state: "Since VIP cells are thought to be dendrite-targeting and display smaller amplitudes than PV cells, we first had to test whether we would be able to detect small VIP cell inputs under these recording conditions." This description does not fully reflect the existing literature. Although VIP interneurons are often characterized as dendrite-targeting, several anatomical and functional studies have demonstrated that they can also innervate the soma of other interneurons. The authors are encouraged to revise this statement to acknowledge these findings and to provide a more balanced overview of the VIP neurons.

Line 256–257: Can the authors clarify why the reported connectivity rate for both PV to SST and VIP to SST motifs differs from those described in Walker et al 2016? It would be helpful if the authors could discuss potential reasons for these discrepancies, such as differences in animal age, slice preparation, recording conditions, or criteria used to define connectivity.

It is unclear how the reciprocal connectivity among PV, SST, and VIP neurons, including SST cells acting as presynaptic partners onto PV and VIP neurons, and VIP cells potentially acting presynaptically onto both PV and SST neurons, might affect the interpretation of the data, given that both pre- and postsynaptic neurons are firing action potentials. The authors should clarify how this experimental design allows them to isolate the functional effect of presynaptic PV or VIP cells on SST cell output under these conditions (1 s stimulation).

Line 270–274: The use of one-second somatic current injections to drive continuous firing seems arbitrary. It is unclear how the number of presynaptic action potentials was determined and whether it varied across cell pairs or experiments. Clarification is needed on how consistent firing rates were and how representative these patterns are of physiological activity.

In Figure 2, PV and VIP cells reduced SST firing rates, but the initial spikes remained largely intact. This suggests SST neurons may still influence downstream targets. The authors should analyze firing rate versus initial spike or discuss how these dynamics affect interpretation of functional connectivity.

Line 312–315. The authors' assertion that the lack of significant differences in spike loss between PV to SST and VIP to SST pairs was unexpected is not entirely accurate when presynaptic neurons are allowed to fire multiple action potentials. VIP and PV interneurons exhibit distinct synaptic properties, which can influence the postsynaptic response.

Reviewer #3

(Remarks to the Author)

The manuscript entitled "Consequences of individual PV and VIP interneuron firing on the output of postsynaptic SST neurons in mouse barrel cortex" by Preuß et al. presents a well-executed study using paired patch-clamp recordings to investigate the inhibitory connections from PV and VIP interneurons onto SST neurons in the mouse barrel cortex. The finding that both PV and VIP cells can significantly reduce action potential output in postsynaptic SST neurons is compelling and challenges the simplified view of interneuron function in cortical circuits. Furthermore, the novel anatomical results from the putative contact site analysis are particularly noteworthy and contribute significantly to the study's impact.

I support the publication of this work; however, I believe a revision is necessary before being suitable for publication in *Communications Biology*.

Major Points:

1. The main limitation of the current manuscript is its insufficient discussion of the functional significance of the findings. Since the main motifs described in this study were previously known, I'd like to ask the authors to discuss in detail what

these new findings add to our broader knowledge of cortical circuit function. A revised manuscript should include a more in-depth discussion on:

- o The potential roles of the current findings of PV→SST and VIP→SST inhibition in shaping the activity of the cortical circuit, particularly in the context of sensory processing.

- o How these findings challenge or refine the classical "disinhibitory" circuit model involving VIP, SST, and PV interneurons.
- o Graphical abstract: A visual summary would greatly enhance the paper. I suggest the authors include a schematic diagram to illustrate their findings and integrate them into a broader circuit model.

2. In the current version of the manuscript, the authors show the connection between PV/VIP cells and SST GABAergic interneurons (IN) with high-frequency firing in presynaptic cells. Did the authors try to induce single action potentials in the presynaptic cells, and if so, did they not see connections with this K-gluconate-based solution, like in a previous paper (Walker et al., 2016)? If not, would they suggest that a single action potential from a VIP cell wouldn't be enough to induce a "relevant" postsynaptic response, like CCK+ MFA INs (Losonczy et al., 2004)? In addition to the exemplar paired recordings shown in Fig. 1G+H, it would be nice to see what the connections looked like when only a single action potential was evoked in the presynaptic cells. Furthermore, the authors could compare the amplitude, rise time, decay time, and half-width of the postsynaptic responses seen in this case, similar to Walker et al., 2016, and to show these results instead of the ones in Fig. S2.

3. Considering the data points included in the statistics, the readers can easily recognize population outliers (e.g., one or two in the case of PV cells and three or four for VIP cells). Specifically, the outliers presented in Fig. 2E+F appear to correspond to the cells described on page 12, line 289. It is noteworthy that the only two and three "significantly important" cells, respectively, are these outliers from the total population of fifteen and twenty-one recorded pairs. This raises a critical question regarding the consistency of these data points across all statistical analyses. Therefore, I would inquire whether the results would remain robust, or if different findings would emerge as statistically significant, upon the exclusion of these extreme outliers from the analysis.

4. In my opinion, the firing of the SST cell shown in Fig. 4A is not the best choice. The cell shows a kind of scattering activity at the end of the one-second-long square pulse. This may confuse readers as to whether the "stop" or scattering occurs due to the inhibition provided by PV cells, or if it's simply an internal active membrane characteristic of the given SST-IN. I would suggest changing this firing with another one similar to Fig. 4B.

Minor Points:

- Page 6, Line 131: Is the pH of the intracellular solution correctly written? The pH of the intracellular solution is typically between 7.2-7.3 to mimic the physiological intracellular pH of a neuron (7.0-7.4). While not incorrect, this value is unusual.
- Page 32, Line 850: The color of the figure legend is different here, please correct it.
- In Fig. 4A+4B, is the Y-axis scale bar incorrect, or were the action potentials of the cells so small?

Version 1:

Reviewer comments:

Reviewer #1

(Remarks to the Author)

The authors have addressed all but one concerns, thank you.

The authors state that "Cell pairs were determined connected when we observed time-locked IPSCs to presynaptic action potentials during our high-frequency firing connection test."

Please allow me to clarify my question: When was a postsynaptic response regarded a time-locked uIPSC and when was it regarded a sIPSC? Please add details to the analysis/definition of synaptic pairs.

Reviewer #2

(Remarks to the Author)

The authors have prepared an excellent revision and we complement them on a beautiful study.

Reviewer #3

(Remarks to the Author)

The authors have adequately addressed all my comments and concerns. Therefore, I recommend that the article be published in Communications Biology.

Referee expertise:

Referee #1: Expert in synaptic physiology

Referee #2: Expert in synaptic physiology

Referee #3: Expert in synaptic physiology

Reviewers' comments:

Reviewer #1 (Remarks to the Author):

The manuscript entitled “Consequences of individual PV and VIP interneuron firing on the output of postsynaptic SST neurons in mouse barrel cortex” by Preuss et al. investigates the innervation and modulatory capacity of PV and/or VIP interneurons on action potential firing in postsynaptic SST interneurons. The authors approach these questions by combining dual recordings between interneuron pairs with morphological reconstructions of recorded pairs.

Overall, the manuscript is well written and the introduction offers a good overview of the current literature regarding interneuronal output control. The study finds high coupling probabilities between pairs of PV-SST and VIP-SST interneurons. In a lot of cases, these connections appear reciprocal. In contrast to the notion that PV interneurons provide somatic inhibition whereas VIP interneurons tend to provide dendritic inhibition, Preuss et al find that both interneuron types exert a comparable output control in SST interneurons. The discussion recapitulates the findings of the present study and integrates these findings into current concepts of interneuronal output control. While the study offers new details on disinhibitory circuit motifs between PV/VIP interneurons and SST interneurons, there are some (mainly methodological) concerns that need to be addressed:

Thank you very much for the very positive assessment of our study. Please find our responses to your criticism below.

- What is the estimated chloride equilibrium potential in the authors’ experimental settings? It is surprising to see somatic (“electrode-seen”) uIPSPs in Fig. 2D at an estimated SST interneuron membrane voltage of -40 mV or less?

The calculated chloride equilibrium potential is ≈ -86 mV. In our experiments we are thus ≈ 40 mV away from it. This results in roughly half the driving force of our earlier study (Walker et al. 2016) where we used a cesium-based solution and a holding potential of 0 mV. In the latter, we often observed VIP IPSCs of around 10 pA maximal amplitude and lower.

- What were the exact parameters according to which a synaptic connection was determined (a) successful or (b) unsuccessful?

To further clarify, a synaptic connection does not include a success rate of 100 % in synaptic transmission. In the Walker et al. study, we observed a synaptic success rate of $\approx 80\%$ in L2/3 VIP to L2/3 SST cells. In other layers or in translaminar connections, we reported even lower success rates (Rachel et al. 2025, Preuss et al. 2025). In the present study, we tested whether cell pairs were synaptically coupled (connected) or not. Cell pairs were determined connected when we observed time-locked IPSCs to presynaptic action potentials during our high-frequency firing connection test.

- Overall, the authors report very high coupling rates between pairs of recorded interneurons. Can the authors discuss/speculate on the reasons of finding such high coupling rates in contrast to other studies (e.g. Jiang et al., 2015, PMID:26612957; Campangola et al., 2022, doi: 10.1126/science.abj5861)?

In our opinion, it makes all the difference whether you are mapping connections without an hypothesis with high throughput approaches or whether you optimize the recording conditions according to a specific scientific question. Thus, we think that without a careful optimization as to distance between the recorded neurons or their relative placement to each other, with respect to known arborization of the axon of the presynaptic cell and dendritic arborization of the postsynaptic cell, one will strongly underestimate the connection probability or even miss it.

- Could the authors provide exemplar recordings of reciprocally connected cell pairs?

We are unsure what you mean by exemplar reciprocally connected pairs. Due to our recording protocols (first SST cells without presynaptic stimulation, then SST cell with presynaptic recording) the condition whether the recorded cells are reciprocally connected or not does not interfere with our analysis.

If you are asking about spike reduction recordings of SST to PV or SST to VIP cells, we only did that in one cell pair. Our recordings often scratched at the 60-minute mark. Therefore, it was not possible to record all the necessary traces to fully characterize the “reverse” SST to PV or SST to VIP connections.

- Given that the authors found +80% of reciprocally connected PV-SST interneuron pairs, it would have been interesting to compare SST interneuron output to either PV and VIP interneuron output. Has this analysis been done in some cells?

No, since we were not able to perform spike reduction recordings of the SST to PV and SST to VIP motif, we did not further analyse the unitary properties of SST to PV and SST to VIP inputs.

- Have the authors considered inherent sweep-to-sweep variability of SST interneurons upon identical current pulse injections in their analysis?

Yes, indeed we considered the sweep-to-sweep variability. Unfortunately, this is something we cannot completely avoid with our experimental setup. If it would have been a major factor, we might have seen an increased SST cell firing in some of the weaker connected pairs (smaller overall IPSCs). This is something that we did not observe. We also believe that variability between beginning and end of such a long recording is a much larger potentially confounding factor. To keep this variability

as small as possible, we alternatingly recorded SST cell firing with and without presynaptic stimulation.

- Could the authors please provide an explanation of why uIPSC amplitudes/latencies are partly divergent from their previous study (Walker et al., 2016, <https://doi.org/10.1038/ncomms13664>)?

There are two answers for why we observed different results in this and the previous study. On the one hand, we used slightly older animals in the present study. In Walker et al. P21 mice were used whereas the present study used mice from P24 on, the vast majority being above P28. On the other hand, we used different internal solutions. Walker et al. used a cesium-based solution in SST cells. Cesium allows to hold SST cells at 0 mV which increases the driving force for inhibitory chloride current. At the same time, cesium also makes the cell electronically more compact, which reduces the space clamp error.

- The authors might want to include a quantitative comparison of phase planes of synaptically delayed action potentials in SST interneurons.

If we understand this correctly, you are asking for a comparison of phase plane plots of SST cell action potentials with and without presynaptic stimulation to analyse the slope of the rising phase and therefore the availability of Na⁺ channels. We are not sure how this analysis would strengthen our main findings of PV and VIP cell induced activity in reduction SST cell activity. Furthermore, we argue that this analysis would be more convincing if we were looking at different target cells.

- The authors should consider a preliminary assessment of postsynaptic GABAergic receptors on SST interneurons as another powerful output modulator.

Thank you for this suggestions, which certainly is a scientifically valuable one. However, since this needs a completely different line of experiments, we consider it as a possible future project.

Reviewer #2 (Remarks to the Author):

While the manuscript addresses inhibitory interactions among interneuron subtypes, the novelty of this observation is somewhat limited, as it is already well established that interneurons can inhibit one another. The strength of the work lies in the effort to dissect specific inhibitory motifs between PV, VIP, and SST neurons; however, unresolved methodological and interpretational concerns substantially limit the overall impact and contribution of the study. Importantly, this study builds directly on the authors' earlier work (Walker et al., 2016) and, while it provides additional detail, it reads more as a continuation of that study rather than a substantial advance beyond it.

Thank you for evaluating our work. We think that there may be a misunderstanding, however. In contrast to nearly all previous studies, to be of our knowledge, we here for the first time assess the influence that a single/individual presynaptic VIP or PV neuron has on its SST target cell output. Previous studies only showed that an IPSP/C of a certain amplitude (often very small) is caused by an AP of a single presynaptic cell. For disinhibition to become effective, this input needs to be transformed in an reduced output, which is not self-evident since GABAergic neurons like SST cells

are not well characterized in terms of their active conductance which could amplify or dampen inputs from presynaptic neurons (of whatever type, here: VIP and PV cells). The few studies that showed a reduced output used optogenetics, which recruits an unknowingly large ensemble of presynaptic cells, leaving the single-cell effect unknown. Therefore, we believe that our study offers an unprecedented advance in understanding inhibitory-inhibitory neuronal interactions at the identified single cell level.

Lines 83–86: The authors state that VIP cells primarily target dendrites. However, previous studies have shown that VIP neurons can also innervate the soma of other interneurons (e.g., Eur J Neurosci. 2007 Apr;25(8):2329–2340. doi: 10.1111/j.1460-9568.2007.05496.x).

Yes, we think that is a good idea to point out the existence of soma-targeting VIP synapses. However, we still think that it is correct to state that VIP cells predominantly target dendrites. To be more transparent we added the following to our discussion:

Lines:533-537

“Nevertheless, there are studies that displayed a strong innervation of PV cell somas by VIP cells (Dávid et al. 2007, Hioki et al. 2013). However, this specificity of VIP synapses is not observed for all cell types, since the overall amount of soma-targeting VIP synapses remains below 10 % in L2/3 of the rodent neocortex (Zhou et al. 2017).”

Line 113, Materials and Methods section: The authors state that experiments were performed on mice of both sexes, postnatal days 24–44 (median: P34). This is inconsistent with other parts of the manuscript. For example, on line 279, it is reported that recordings and analyses of 15 PV → SST pairs were performed in mice aged P25–P48 (median: P42, from 14 animals).

Thank you for making us aware of this discrepancy. Yes, you are correct. The overall age of all used mice was incorrect and has been corrected now (line 113): postnatal day 24-55 (median 35)

Line 209: Morphological reconstruction and putative contact analysis. Although Airyscan imaging improves upon standard confocal resolution, the method as described is not sufficient to confirm synaptic connectivity. Identifying putative contact sites based solely on gapless apposition risks a high false-positive rate, as axons and dendrites frequently pass in proximity without forming synapses. The lack of systematic validation with synaptic markers or electron microscopy makes it difficult to assess the accuracy of the reported PCS. Consequently, the conclusions drawn from these counts should be interpreted with caution until supported by orthogonal approaches.

Yes, you are correct that other techniques should be used to undoubtedly quantify synaptic connections. Therefore, we decided to use the term *putative contact site* and mostly refrained from using the term *synapse*. Unfortunately, we do not have the opportunity to use electron microscopy. Also our mouse lines did not allow any further validation of synaptic markers. SST cells are genetically labelled with GFP and VIP-cells with TdTomato. In addition, we used DAPI staining, and Alexa 633 for visualization of analysed cell pairs. Therefore, we have no possibility to use any further markers using classical fluorescence microscopy without the risk of channel-crosstalk.

Line 242-244: In a previous study, we were able to show that in triple transgenic animals, GIN cells in L2/3 can all be classified as MC, which are targeted by PV and VIP cells. The reference is missing.

We added the Walker et. al. 2016 reference

Line 249–251: The author's state: "Since VIP cells are thought to be dendrite-targeting and display smaller amplitudes than PV cells, we first had to test whether we would be able to detect small VIP cell inputs under these recording conditions." This description does not fully reflect the existing literature. Although VIP interneurons are often characterized as dendrite-targeting, several anatomical and functional studies have demonstrated that they can also innervate the soma of other interneurons. The authors are encouraged to revise this statement to acknowledge these findings and to provide a more balanced overview of the VIP neurons.

Yes, you are correct. As previously mentioned, the VIP also target somas of interneurons although this has been shown best for VIP synapses on PV cells. Nevertheless, the statement that they predominantly target-dendrites is not false. In addition, the amplitude size is the more critical point in this statement. We changed it to the following phrase:

" Since most frequently VIP neurons target dendrites and not somata (Zhou et al., 2017), we assumed low-amplitude inputs of VIP neurons to SST cells. Thus, we first had to ..."

Line 256-257: Can the authors clarify why the reported connectivity rate for both PV to SST and VIP to SST motifs differs from those described in Walker et al 2016? It would be helpful if the authors could discuss potential reasons for these discrepancies, such as differences in animal age, slice preparation, recording conditions, or criteria used to define connectivity.

We suspect that the age of animals used in our study plays a role for reduced connectivity observed in the PV to SST cell pairs as it has been shown that substantial synaptic pruning of PV synapses takes place in the first weeks of development (Micheva et al 2012 <https://doi.org/10.1523/JNEUROSCI.0871-21.2021>). In Walker et al., age range for PV to SST pairs was P21-36, whereas the median age for connected PV to SST pairs in the present study is P42 (not to confuse with the median age of all used animals P35).

For the VIP motif, we observed an increase in connectivity of ≈ 8 percentage points. We consider that this is either biological variability or due to the difference in connectivity test that was used in the present study. Here we used high frequency firing of presynaptic cells, which led to synaptic facilitation. In Walker et al., single action potentials were used as connection test. The synaptic facilitation might have helped to detect synaptic connections, which would have stayed undetected using only single responses

It is unclear how the reciprocal connectivity among PV, SST, and VIP neurons, including SST cells acting as presynaptic partners onto PV and VIP neurons, and VIP cells potentially acting presynaptically onto both PV and SST neurons, might affect the interpretation of the data, given that

both pre- and postsynaptic neurons are firing action potentials. The authors should clarify how this experimental design allows them to isolate the functional effect of presynaptic PV or VIP cells on SST cell output under these conditions (1 s stimulation).

The good thing about our recording approach is that the influence of the reverse connection (SST to PV/ SST to VIP) has only a minor effect. At first we are recording SST cell firing without presynaptic stimulation and count the number of SST cell action potentials. In the next step we also activate the presynaptic cell and count SST and presynaptic spikes (from PV or VIP cells). If the cell pairs are reciprocally connected, we might observe a slightly reduced number of action potentials in the presynaptic cells in comparison to the square pulse at double or triple rheobase without SST cell activation. The fact that the number of presynaptic action potentials is slightly reduced leads to a small underestimation of effect size when looking at the general effect strength (like in Fig 2E or 4C). Since we additionally calculate the effect strength normalized for the number of presynaptic spikes, we also report an effect strength where this "error" is taken out of the equation.

Line 270-274: The use of one-second somatic current injections to drive continuous firing seems arbitrary. It is unclear how the number of presynaptic action potentials was determined and whether it varied across cell pairs or experiments. Clarification is needed on how consistent firing rates were and how representative these patterns are of physiological activity.

In Figure 2, PV and VIP cells reduced SST firing rates, but the initial spikes remained largely intact. This suggests SST neurons may still influence downstream targets. The authors should analyze firing rate versus initial spike or discuss how these dynamics affect interpretation of functional connectivity.

We used 1s-long square pulses as this is the classical way of characterizing individual neurons in acute slices. Since SST cells are characterized by an continuous adapting firing pattern, we hypothesized that 1 s stimuli are an adequate means to induce firing that differs throughout the full length of the stimulus. As described in the methods, we did not aim for a fixed number of action potentials in pre- and postsynaptic cells but to multitudes of rheobase, the smallest current in which one action potential is evoked. This allowed us to evoke continuous firing with little variety between sweeps. Therefore, we observed differences in the overall number of action potentials, since it was determined by the internal properties of each individual cell. We are aware that square pulses are not ideal to mimic physiological firing of neurons in vivo. Dynamic clamp amplifiers might be a more powerful tool for that purpose but were not available in our lab.

Nevertheless, all cell types, including SST cells, display long spontaneous or evoked phases of spiking activity in vivo (as seen in Kiritani et al 2023, <https://doi.org/10.1371/journal.pone.0287174>). Our firing rates are either at the lower edge or above physiological firing rates for VIP and SST cells. PV cell firing was above physiological levels. However, lower currents (2x the rheobase) were not possible as this would have resulted in stuttering firing and fine tuning of current levels would have become arbitrary on a cell to cell basis.

To the second part: Yes, it is correct that the first spike remains largely intact. There are two things that we would like to point out. On the one hand, we are only looking at a single neuron that nevertheless is already able to delay the first spike. We assume that a certain population of VIP cells, being active at the same time, could completely silence the cell. On the other hand, we argue that

this initial action potential might not be the physiologically most relevant spike. Since it has been shown that all cell types show long windows of consecutive firing, the second or third frame (named "mid" or "late") might be more relevant. In these time frames we observed complete silencing by just a single presynaptic neuron.

Line 312-315. The authors' assertion that the lack of significant differences in spike loss between PV to SST and VIP to SST pairs was unexpected is not entirely accurate when presynaptic neurons are allowed to fire multiple action potentials. VIP and PV interneurons exhibit distinct synaptic properties, which can influence the postsynaptic response.

We agree that one cannot simply compare unitary synaptic properties when presynaptic neurons are firing multiple action potentials. Nevertheless, when we designed our experiments, we did not expect that short-term plasticity would be able to level out the large differences of amplitude size that we observed in Walker et al 2016 and in the present study. Therefore, it was unexpected for us at first.

Reviewer #3 (Remarks to the Author):

The manuscript entitled "Consequences of individual PV and VIP interneuron firing on the output of postsynaptic SST neurons in mouse barrel cortex" by Preuß et al. presents a well-executed study using paired patch-clamp recordings to investigate the inhibitory connections from PV and VIP interneurons onto SST neurons in the mouse barrel cortex. The finding that both PV and VIP cells can significantly reduce action potential output in postsynaptic SST neurons is compelling and challenges the simplified view of interneuron function in cortical circuits. Furthermore, the novel anatomical results from the putative contact site analysis are particularly noteworthy and contribute significantly to the study's impact.

I support the publication of this work; however, I believe a revision is necessary before being suitable for publication in Communications Biology.

We are very happy about the very favourable review of our study and thank you for that. We have spent all efforts possible to improve the manuscript according to your suggestions, without becoming too speculative.

Major Points:

1. The main limitation of the current manuscript is its insufficient discussion of the functional significance of the findings. Since the main motifs described in this study were previously known, I'd like to ask the authors to discuss in detail what these new findings add to our broader knowledge of cortical circuit function. A revised manuscript should include a more in-depth discussion on:

- o The potential roles of the current findings of PV→SST and VIP→SST inhibition in shaping the activity of the cortical circuit, particularly in the context of sensory processing.

- o How these findings challenge or refine the classical "disinhibitory" circuit model involving VIP, SST, and PV interneurons.

o Graphical abstract: A visual summary would greatly enhance the paper. I suggest the authors include a schematic diagram to illustrate their findings and integrate them into a broader circuit model.

As requested, we added the following part to our discussion (line 560-577) :

Distinct roles of PV and VIP Interneurons in information processing

The PV interneuron subpopulation can be considered the “all-rounder” and the primary workhorse in all existing forms of inhibition, including disinhibition (Tremblay et al. 2016, Feldmeyer et al. 2018, Preuss et al. 2025). During bouts of foveal whisking, in order to achieve decision making in a tactile discrimination paradigm, the PV cell-mediated inhibition/disinhibition circuit motifs may be tuned to achieve fast evidence accumulation and consequently correct decision making (Khilkevich et al. 2024) due to the high inhibitory current flow right from the start of their firing. By contrast, disinhibition more than inhibition may dominate VIP cell-mediated effects and, due to initially weak but then increasing inhibitory current flow, with some delay, they may open input channels (Williams et al. 2025) that are necessary for more refined and extended evidence accumulation during longer bouts of foveal whisking before a decision can be made.

Our thinking on disinhibition has to be refined in the sense that every GABAergic interneuron subpopulation (also the ones not mentioned here, like somatostatin-expressing neurons or LAMP5/NDNF-expressing neurogliaform cells) seems to be involved -to a certain extent- in disinhibitory circuit motifs (Lee et al. 2015, Donato et al. 2023). So the question in the future should not be “Can a certain type of IN perform disinhibition?” but when and how does it do it.

2. In the current version of the manuscript, the authors show the connection between PV/VIP cells and SST GABAergic interneurons (IN) with high-frequency firing in presynaptic cells. Did the authors try to induce single action potentials in the presynaptic cells, and if so, did they not see connections with this K-gluconate-based solution, like in a previous paper (Walker et al., 2016)? If not, would they suggest that a single action potential from a VIP cell wouldn't be enough to induce a "relevant" postsynaptic response, like CCK+ MFA INs (Losonczy et al., 2004)? In addition to the exemplar paired recordings shown in Fig. 1G+H, it would be nice to see what the connections looked like when only a single action potential was evoked in the presynaptic cells. Furthermore, the authors could compare the amplitude, rise time, decay time, and half-width of the postsynaptic responses seen in this case, similar to Walker et al., 2016, and to show these results instead of the ones in Fig. S2.

We did not perform single action potential experiments, as this would have been mostly a reproduction of our previous Walker et al. study (with yet another internal solution) and probably would not have been of major interest to the readership. In the present study, we wanted to focus onto the functional consequences of the inhibitory inputs on the spiking output of the SST cell and not perform a detailed characterization of unitary connection properties. Our recordings often reached or overshot the 60 min mark. Therefore, we did not have the time to perform a substantial amount of high-quality single spike recordings in addition to the experiments we really wanted to do.

We doubt that a single VIP spike can have a “relevant” effect on a postsynaptic cell in most cases. Nevertheless, in contrast to Losonczy et al. 2004, we did not need multiple tens of action potentials to obtain clear responses (IPSCs).

Unfortunately, we cannot analyse the unitary properties any further than we currently show. The reason is that with our high frequency firing approach to improve the detectability of a unitary connection, we recorded superimposed IPSCs. This makes it impossible to precisely analyse parameters as decay time and half-width. This would have required single action potentials which would have been too time intensive and mostly a reproduction of our previous work.

3. Considering the data points included in the statistics, the readers can easily recognize population outliers (e.g., one or two in the case of PV cells and three or four for VIP cells). Specifically, the outliers presented in Fig. 2E+F appear to correspond to the cells described on page 12, line 289. It is noteworthy that the only two and three “significantly important” cells, respectively, are these outliers from the total population of fifteen and twenty-one recorded pairs. This raises a critical question regarding the consistency of these data points across all statistical analyses. Therefore, I would inquire whether the results would remain robust, or if different findings would emerge as statistically significant, upon the exclusion of these extreme outliers from the analysis.

Yes, you are mentioning a point that we have been discussing intensively in the lab. Nevertheless, we would not call them outliers, since in all paired recordings we and others performed such unusually strong connections showed up. In LeFort et al. 2009, they were computationally modelled and considered to be an important feature and not a bug in the network. Thus, there is no scientific reason to exclude these examples from our data set. These are sound recordings of PV to SST or VIP to SST cell pairs. As you pointed out correctly, we also mentioned these examples in the text. We also used the median which is less prone to be influenced by these examples than for example the mean. As an example: if we would discard the cell pairs with SST cell spike reduction >30 % (although there is no scientific/statistical reason) the PV to SST median of firing reduction goes from 11.92 % to 11.04 %. The same is true for VIP to SST: from 5.92 % to 5.05 %.

4. In my opinion, the firing of the SST cell shown in Fig. 4A is not the best choice. The cell shows a kind of scattering activity at the end of the one-second-long square pulse. This may confuse readers as to whether the “stop” or scattering occurs due to the inhibition provided by PV cells, or if it's simply an internal active membrane characteristic of the given SST-IN. I would suggest changing this firing with another one similar to Fig. 4B.

After very long recording times, we often observed scattering firing in our (continuously adapting) SST cells. This excessively happened at the later time points of evoked SST cell firing and was also observed without presynaptic stimulation. Therefore, the scattering occurred independently from presynaptic stimulation. To not disturb our data sets by this, we were only focussing on the early time windows of SST cell firing for quantitative analysis. Nevertheless, we understand that this could confuse the reader and therefore changed the PV to SST example to another one where there was no scattering.

Minor Points:

- Page 6, Line 131: Is the pH of the intracellular solution correctly written? The pH of the intracellular solution is typically between 7.2-7.3 to mimic the physiological intracellular pH of a neuron (7.0-7.4). While not incorrect, this value is unusual.

Yes, the intracellular solution we have been using for years now has a pH 7.4, which is still in the range of a neurons pH.

- Page 32, Line 850: The color of the figure legend is different here, please correct it.

Corrected!

- In Fig. 4A+4B, is the Y-axis scale bar incorrect, or were the action potentials of the cells so small?

You are correct. The scale bar is supposed to be 20 mV and not 10 mV. It has been changed in the revised version.

We would like to thank all reviewers for their critical comments and valuable feedback! We are sorry that we did not answer Reviewer 1 comments to the fullest. We would like to address the last remaining question with these comments.

REVIEWERS' COMMENTS:

Reviewer #1 (Remarks to the Author):

The authors have addressed all but one concerns, thank you.

The authors state that "Cell pairs were determined connected when we observed time-locked IPSCs to presynaptic action potentials during our high-frequency firing connection test."

Please allow me to clarify my question: When was a postsynaptic response regarded a time-locked uIPSC and when was it regarded a sIPSC? Please add details to the analysis/definition of synaptic pairs.

Thank you for your clarification. Time-locked responses were defined as IPSCs with latencies (time from spike peak to IPSC onset) below 3 ms. We have successfully used this definition in previous studies (Walker et al 2016, Rachel et al 2025, Preuss 2026). For analysis, successful IPSCs of the n^{th} presynaptic action potential were aligned to spike peak. If we observed multiple IPSCs with latencies <3ms throughout the 200 ms connection test, these cell pairs were counted as connected pairs.

We added the following sentences to the method part: "We defined neuron pairs as connected when we observed multiple time-locked IPSCs with latencies (time from presynaptic spike peak to IPSCs onset) below 3 ms during the 200 ms high frequency firing connection test."

Reviewer #2 (Remarks to the Author):

The authors have prepared an excellent revision and we complement them on a beautiful study.

Reviewer #3 (Remarks to the Author):

The authors have adequately addressed all my comments and concerns. Therefore, I recommend that the article be published in Communications Biology.